# Circulating miRNA 122-5p Expression Predicts Mortality and Cardiovascular Events in Chronic Hemodialysis Patients: A Multicentric, Pilot, Prospective Study

**DOI:** 10.3390/biom13111663

**Published:** 2023-11-17

**Authors:** Anila Duni, Marta Greco, Pierangela Presta, Roberta Arena, Ethymios Pappas, Lampros Lakkas, Katerina K. Naka, Antonio Brunetti, Daniela Patrizia Foti, Michele Andreucci, Giuseppe Coppolino, Evangelia Dounousi, Davide Bolignano

**Affiliations:** 1Department of Nephrology, School of Medicine, University of Ioannina, 45110 Ioannina, Greece; 2Department of Health Sciences, Magna Graecia University, 88100 Catanzaro, Italy; 3Clinical Pathology Lab., Magna Graecia University, 88100 Catanzaro, Italy; 4Nephrology and Dialysis Unit, Magna Graecia University, 88100 Catanzaro, Italy; 5Hemodialysis Unit, General Hospital of Filiates, 46300 Filiates, Greece; 6Second Department of Cardiology, University Hospital of Ioannina, 45500 Ioannina, Greece; 7Department of Experimental and Clinical Medicine, Magna Graecia University, 88100 Catanzaro, Italy; 8Department of Medical and Surgical Sciences, Magna Graecia University, 88100 Catanzaro, Italy

**Keywords:** miRNA 122-5p, hemodialysis, cardiovascular risk, biomarker

## Abstract

Background: Despite patients undergoing chronic hemodialysis (HD) being notoriously prone to adverse cardiovascular (CV) events, risk prediction in this population remains challenging. miRNA 122-5p, a short, non-coding RNA predominantly involved in lipid and carbohydrate metabolism, has recently been related to the onset and progression of CV disease. Methods: We run a pilot, multicenter, longitudinal, observational study to evaluate the clinical significance and prognostic usefulness of circulating miRNA 122-5p in a multicentric cohort of 74 individuals on maintenance HD. Results: Patients displayed lower circulating miRNA 122-5p as compared to healthy controls (*p* = 0.004). At correlation analyses, ALT (β = 0.333; *p* = 0.02), E/e’ (β = 0.265; *p* = 0.02) and CRP (β = −0.219; *p* = 0.041) were independent predictors of miRNA 122-5p levels. During a median follow-up of 22 months (range of 1–24), 30 subjects (40.5%) experienced a composite endpoint of all-cause mortality and fatal/non-fatal CV events. Baseline circulating miRNA 122-5p was higher in these subjects (*p* = 0.01) and it predicted a significantly higher risk of endpoint occurrence (Kaplan–Meier crude HR 3.192; 95% CI 1.529–6.663; *p* = 0.002; Cox regression adjusted HR 1.115; 95% CI 1.009–1.232; *p* = 0.03). Conclusions: Altered miRNA 122-5p levels in HD patients may reflect hepatic and CV damage and may impart important prognostic information for improving CV risk prediction in this particular setting.

## 1. Introduction

Individuals with end-stage kidney disease (ESKD) undergoing chronic hemodialysis carry a substantial burden of cardiovascular (CV) disease, with CV mortality accounting for nearly half of the deaths in this group of patients [1,2]. The pathogenesis of CV disease in ESKD is complex and relies on the interplay between classical and non-traditional ESKD-related risk factors, such as uremic toxins, inflammation, enhanced oxidative stress and many others that still remain unknown [3]. Consequently, CV risk prediction is challenging in the majority of ESKD patients and the identification of new biomarkers able to improve such prediction is nowadays considered a research priority.

miRNA 122-5p is a short, non-coding RNA mostly expressed in the liver, which is predominantly involved in the regulation of lipid and carbohydrate metabolism [4,5]. Altered miRNA 122-5p levels have been found in the setting of drug-induced hepatocyte toxicity and viral hepatitis, as well as in systemic diseases leading to liver failure, thereby serving as a potential indicator of disease severity [6]. However, under either physiological or pathological conditions, miRNA 122-5p can also target various peripheral organs and tissues [7]. At the CV level, a deranged miRNA 122-5p expression exerts detrimental effects on atherosclerosis progression and cardiomyocytes survival [8,9], may aggravate cardiac and vascular fibrosis in the presence of renal damage [10] and may accelerate cardiac hypertrophy progression in rat models of diabetic cardiomyopathy [11].

Interestingly, deranged circulating miRNA 122-5p levels predict worse outcomes in either experimental or clinical ischemic cardiac injury [12] and can improve CV risk stratification in patients with myocardial infarction and adverse ventricular remodeling in the long term [13].

Yet, despite a growing body of evidence that is nowadays pointing at miRNA 122-5p as a potential mediator and biomarker for CV disease, to date, very scant evidence is available for the high-risk setting of ESKD [14]. Conversely, in a previous study, we demonstrated that three other miRNAs (30a-5p, 23a-3p and 451a) putatively involved in CV disease may serve as sensitive biomarkers of mortality and CV events among chronic hemodialysis patients [15].

Keeping in mind such a background, we have therefore designed a pilot longitudinal study to investigate, for the first time, the possible clinical significance and prognostic usefulness of circulating miRNA 122-5p in a multicentric cohort of ESKD individuals on maintenance hemodialysis.

## 2. Materials and Methods

### 2.1. Study Design and Participants’ Selection Criteria

We ran an observational, prospective, multicenter cohort study on 74 chronic HD patients undergoing regular dialysis at the “Mater-Domini” University hospital of Catanzaro, Italy (*n* = 28), the University Hospital of Ioannina, Greece (*n* = 23) and the General Hospital of Filiates, Greece (*n* = 23). All patients followed a typical three times/week dialysis scheme, by standard bicarbonate hemodialysis or hemodiafiltration, displayed an unchanged therapeutic scheme for at least 3 months before study start and had achieved a stable dry-weight and a normotensive edema-free state. Exclusion criteria were the following: acute or non-intermittent hemodialysis, dialysis vintage < 6 months, switch from peritoneal dialysis or renal transplantation, recent hospitalization (<1 month) for CV events, malignancy, liver, thyroid or infectious diseases, alterations in their white cell count or current treatment with steroids or immunosuppressors. The local Ethic Committees approved the study and fully informed consent was obtained from all participants.

### 2.2. Clinical and Laboratory Assessment

Clinical, laboratory, anthropometric and dialysis parameters were recorded before starting a mid-week dialysis session. Biochemical data were quantified following the standard methods used in the routine clinical laboratory. The natural logarithm of the ratio between initial and final urea concentration (Kt/V) was considered to estimate dialysis adequacy. Blood pressure was measured three times consecutively before dialysis start and the average value was considered for data analysis. Additionally, all patients underwent an echocardiographic measurement, performed by an expert operator who was unaware of the patient’s clinical status. The principal parameters for estimating ventricular function and morphology, and for cardiac chamber quantification, were assessed as recommended elsewhere [16].

### 2.3. miRNA 122-5p Extraction and Quantification of Relative Expression

Blood samples were collected before starting the middle-week dialysis session. Biochemical parameters were measured in all patients by Cobas 8000 (Roche Diagnostics, Basel, Switerland) using the relative kits (Roche Diagnostics, Basel, Switzerland). Blood count analysis (Hb, RBC, WBC and platelet counts) was performed using ADVIA 2120i (Siemens Healthcare Diagnostics, Marburg, Germany). Fibrinogen was determined using BCS XP (Siemens, Healthcare Diagnostics, Marburg, Germany) using the Clauss method. All the above-mentioned assays were carried out according to the manufacturers’ instructions. Further blood samples were centrifuged at 3000 rpm for 10–15 min at 4 °C and aliquots were stored at −80 °C before being further processed to remove insoluble material. Circulating miRNAs were extracted from 200 µL of serum from both HD patients and healthy controls using a commercial column-based system (miRNeasy Serum/Plasma Advanced Kit, Qiagen, Germany) to enrich low-molecular-weight RNA fractions (LMW RNA). Extracted RNA was then eluted in 20 µL of RNase-free water, according to the manufacturer’s protocol. The level of hemolysis in serum samples was determined spectrophotometrically at 414 nm. Five ng/μL of the LMW RNA fraction were employed for reverse transcription with miRCURY LNA-RT Kit (Qiagen, Germany) using a thermocycler system (GeneAmp 2700 Thermal Cycler, Applied Biosystems, MA, USA). UniSp2, 4 and 5 spike-in mix (Qiagen, Germany) were used to monitor RNA extraction and reverse transcription efficiency. The reverse transcription reaction was then eluted with water (1:40), and 4 µL were employed for quantitative RT-PCR (RT-qPCR) using miRCURY LNA SYBR Green PCR Kits (Qiagen, Germany) and the CFX96 Real-Time PCR System (Bio-Rad Laboratories, Hercules, CA, USA). For RT-PCR, Focus PCR Panels (Qiagen, Germany) containing, in 96-well plates, miRNA-specific forward and reverse primers for the miRNA 122-5p reference gene, control assays (UniSp6, UniSp2, UniSp3, UniSp4) and endogenous controls (miR-486-5p) were used. The UniSp6 RNA spike-in was employed as a reverse transcription positive control and as an inter-plate calibrator, according to the manufacturer’s instructions. Relative miRNA expression was evaluated using the 2^−ΔCt^ method for each sample [17]. ΔCt values were calculated as the raw Ct value − raw Ct value for the selected reference miRNA (ΔCt = Ct [miRNA122-5p] − Ct [miR-486-5p]). All analyses were performed in the same laboratory (Clinical Pathology Lab, “Mater-Domini” University hospital, Catanzaro, Italy) and were made blind. miRNA 122-5p expression was also evaluated in a small group of 14 healthy matched subjects (*n* = 10 male) with no evidence or clinical history of renal, hepatic or CV disease.

### 2.4. Prospective Follow-Up and Study Endpoint

After the baseline evaluation, patients were prospectively followed for 24 months or until the occurrence of a combined endpoint of all-cause mortality, CV mortality and non-fatal CV events requiring hospitalization; these latter included coronary, cerebrovascular or peripheral artery disease events, acute heart decompensation or severe cardiac arrythmia.

### 2.5. Statistical Analysis

The statistical analysis was performed using the SPSS package (version 24.0.0.0; IBM corporation, Armonk, NY, USA), the MedCalc Statistical Software (version 14.8.1; MedCalc Software bvba, Ostend, Belgium) and the GraphPad Prism software (version 9.0.0; GraphPad Software LLC, San Diego, CA, USA). The unpaired *t*-test, the Mann–Whitney U test and the chi-square, followed by a Fisher’s exact test, were employed for evaluating differences between study subgroups, as appropriate. Correlation analyses were tested by the Pearson (R) or the Spearman (Rho) coefficients. Data showing a skewed distribution were log-transformed to approximate a normal distribution before being tested for correlations. Multiple regression analyses were performed, considering miRNA 122-5p as the dependent variable and including in the model all the univariate correlates. The diagnostic capacity of miRNA 122-5p to identify patients experiencing the endpoint was tested by receiver operating characteristics (ROC) analyses and the best expression threshold was computed by the Youden index. Kaplan–Meier curves were then generated for patients with miRNA 122-5p expression above or below the optimal ROC-derived cut-off and compared by a log-rank test. Univariate analyses, followed by multivariate Cox proportional hazard regression analyses, were performed to evaluate time-dependent associations with the composite endpoint for variables, which differed at baseline among study subgroups. All endpoint analyses were conducted on a time-to-first event basis. All results were considered significant if the *p* value was ≤0.05.

## 3. Results

### 3.1. Characteristics of the Study Population

Table 1 and Table 2 show the main characteristics of the study population. The mean age was 72.5 ± 12.5 years and the majority of individuals were male (75.7%). The median dialysis vintage was 35.5 months (IQR 17–68.2), with an optimal dialysis adequacy on average (mean Kt/V: 1.44 ± 0.27). More than a half of HD patients (55.4%) followed a standard bicarbonate dialysis regimen while the remainder were on hemodiafiltration. The prevalence of diabetes was 27% (*n* = 20) and 48.6% of individuals (*n* = 36) had a history of any previous CV disease.

### 3.2. miRNA 122-5p Expression in Hemodialysis Patients

Collectively, HD patients displayed a lower normalized relative expression of miRNA 122-5p, compared to healthy controls (0.051 [0.019–0.083] vs. 0.111 [0.050–0.233] 2^−ΔCT^; *p* = 0.004; Figure 1) with an average fold regulation (FR) of −2.42 (*p* = 0.005).

In the univariate analyses, miRNA 122-5p expression was directly associated with ALT (R = 0.450; *p* < 0.001), AST (R = 0.389; *p* = 0.001), E/e’ (R = 0.0291; *p* = 0.01) and LVMi (R = 0.272; *p* = 0.02), while inverse correlations were found with CRP (R = −0.246; *p* = 0.03), serum potassium (R = −0.273; *p* = 0.01) and urea levels (R = −0.257; *p* = 0.02). However, in a fully adjusted multivariate model, including all univariate correlates as independent variables, only the associations with ALT (β = 0.333; *p* = 0.02), E/e’ (β = 0.265; *p* = 0.02) and CRP (β = −0.219; *p* = 0.041) remained significant, while those with LVMi, serum potassium, AST and urea were lost. This multivariate model was remarkably robust, explaining almost 40% of the overall variability (R^2^ = 0.398; *p* < 0.001) of miRNA 122-5p in this cohort.

Table 3 and Figure 2 summarize the findings from the correlation analyses of miRNA 122-5p.

### 3.3. Composite Endpoint during the Follow-Up Period

During a median follow-up of 22 months (range of 1–24), 30 subjects (40.5%) experienced the composite endpoint. In detail, 13 subjects died due to a CV event (*n* = six) or other causes (*n* = seven) while 17 subjects experienced a non-fatal CV event requiring hospitalization (eight coronary events, five severe arrhythmia episodes, four cerebrovascular disease).

At baseline, patients reaching the composite endpoint were older (*p* = 0.04) and had higher levels of troponin I (*p* = 0.04), CK-MB (*p* = 0.02) and LDL cholesterol (*p* = 0.05) but lower diastolic (*p* = 0.04) and pulse pressure values (*p* = 0.03). Echocardiography also revealed a more increased left atrial volume (*p* = 0.05), left ventricular mass (*p* = 0.05) and a more severe diastolic dysfunction, as indicated by higher E/e’ values (*p* = 0.01). A barely significant difference was present in the prevalence of CV diseases (*p* = 0.08) and the percentage of patients treated by hemodiafiltration (*p* = 0.06). No further differences re-merged for the other variables recorded (*p* ≥ 0.10).

Differences in clinical data in patients dichotomized according to the occurrence of the composite endpoint are displayed in Table 1 and Table 2.

### 3.4. Diagnostic and Prognostic Capacity of Circulating miRNA 122-5p Expression

At baseline, patients experiencing the endpoint displayed a significantly increased normalized relative expression of miRNA 122-5p, compared to others (0.071 [0.022–0.106] vs. 0.034 [0.019–0.061] 2^−ΔCT^; *p* = 0.01) (Table 1 and Figure 3). ROC analyses evidenced a significant diagnostic capacity of miRNA 122-5p in identifying patients experiencing the combined endpoint with and AUC of 0.679 (95% CI 0.559–0.783) and a best cut-off value of circulating expression ≥0.0521 2^−ΔCT^ (sensitivity 70%, 95% CI 50.6–85.3; specificity 69.77%, 95% CI 53.9–82.8; Figure 4).

Kaplan–Meier survival curves of event-free patients, according to baseline miRNA 122-5p expression below or above the ROC-derived cut-off value, are presented in Figure 5. Patients with miRNA 122-5p levels above the threshold experienced a significantly faster progression to the combined endpoint (crude HR 3.192; 95% CI 1.529–6.663; *p* = 0.002; log-rank test, χ^2^ = 9.55) as compared to others.

### 3.5. Cox Regression Analyses for the Composite Endpoint

Clinical variables which differed at baseline between the two subgroups were tested by Cox regression analysis to confirm associations with the combined endpoint (Table 4).

At univariate analyses, pulse pressure (HR 1.026; 95% CI 1.008–1.044; *p* = 0.005), E/e’ (HR 1.155; 95% CI 1.081–1.234; *p* < 0.001), CK-MB (HR 1.062; 95% CI 1.014–1.112; *p* = 0.01) and miRNA 122-5p (HR 3.235; 95% CI 1.343–4.805; *p* = 0.03) were confirmed as significant predictors of the established outcome. Conversely, such a correlation was not evidenced for age, LDL cholesterol, troponin I, LAVi and LVMi (*p* ranging from 0.09 to 0.34).

A multivariate Cox model including these univariate predictors reported an independent 11% increase in the risk of the combined outcome per each 2^−ΔCT^ (×10^3^) increase in the circulating miRNA 122-5p relative expression (*p* = 0.03). Also, the severity of diastolic dysfunction remained independently associated with the endpoint (E/e’ HR 1.085; 95% CI 1.028–1.146; *p* = 0.003) while the correlations with pulse pressure and CK-MB were lost.

## 4. Discussion

Two main findings from our study deserve, to our opinion, a focused discussion.

First, we have found reduced circulating miRNA122-5p levels in chronic HD patients as compared to healthy individuals.

However, rather surprisingly, in these patients a higher circulating miRNA122-5p expression held a remarkable association with worsened CV outcomes, thereby indicating a prognostic usefulness of measuring this miRNA in the complex ESKD setting.

Reduced levels of various circulating miRNAs have already been reported in patients with ESKD, undergoing chronic renal replacement therapy [14,15,18,19]. Several plausible mechanisms underlying such a reduction have been suggested, including a substantial removal by the dialysis procedure itself, an augmented enzymatic digestion due to an increased RNAse activity in ESKD or an excessive hemodilution [10,18]. Interestingly, in our study, the circulating levels of miRNA 122-5p were directly associated with ALT, a finding which is consistent with the predominant hepatic origin of this miRNA [4,5]. Taking into consideration that the concentrations of serum aminotransferases in dialysis patients are usually reduced due to pyridoxine deficiency, hemodilution and the presence of inhibitory uremic toxins [20,21], the observed reduction in miRNA 122-5p levels could, in principle, share the same pathophysiological mechanisms.

Of note, in our cohort, circulating miRNA 122-5p levels were also independently associated with increased E/e’ and lower C-reactive protein values. Elevated E/e’ represents an adverse prognostic marker in heart failure as it reflects the severity of left ventricular stiffness, myocardial fibrosis and, ultimately, diastolic dysfunction [22,23]. Despite, in this view, the fact that higher circulating miRNA 122-5p might reflect in some way the hepatic congestion in the case of uremic heart failure, the exact relationships between this miRNA and the severity of cardiac abnormalities remains largely too complex to explain.

In our HD cohort, we also reported a direct univariate association between miRNA 122-5p circulating levels and LVMi, but such a correlation was lost in the fully adjusted multivariate regression models, suggesting the presence of relevant confounding. Previous experimental data have elucidated the role of miRNA 122-5p overexpression in worsening myocardial hypertrophy and the severity of heart failure in hypertensive rats [24] while, in clinical studies, miRNA 122-5p expression is upregulated in hypertensive subjects and correlates with ventricular mass and cardiac dysfunction [25,26]. Since arterial hypertension and left ventricular hypertrophy are both hallmarks of ESKD, future research is advocated to shed light on the exact role of miRNA 122-5p in the complex pathological CV remodeling which characterizes the ESKD setting.

On the contrary, the possible involvement of miRNA 122-5p in the modulation of inflammation is less controversial. In vivo and in vitro models of non-alcoholic fatty liver disease pointed to this miRNA as a potential mediator of hepatic inflammatory and oxidative stress response [27]. Likewise, in cellular models, miRNA 122-5p elicited the inflammatory response by triggering the production and cell-release of IL-1β, IL-6, monocyte chemoattractant protein 1 (MCP-1) and TNF-α [28]. Accordingly, inflammation seems to be strictly linked to circulating miRNA 122-5p levels in our study cohort also, but the biological significance of the inverse relationship found with C-reactive protein—which shares with miRNA 122-5p the same hepatic origin—would deserve further targeted experimental investigations.

Of note, despite the fully adjusted multivariate regression model results being significantly robust (R^2^ = 40%), a large amount of the overall variability of this miRNA in our cohort remained unexplained, thereby suggesting that the biological mechanisms unbalancing miRNA 122-5p expression in HD patients are largely more intricate than those above-postulated.

Nevertheless, in our opinion, the most relevant finding from our study is the strong prognostic capacity of circulating miRNA 122-5p to predict, in these individuals, the occurrence of a combined CV endpoint in the short-mid-term. Such an observation pairs well with previous studies of patients with acute coronary syndromes [12,29] and echoes a recent report from our group in which we demonstrated that a small panel of other miRNAs (30a-5p, 23a-3p and 451a) involved in abnormal cardiac remodeling may hold important prognostic value for CV risk stratification in ESKD patients [15].

In the present study, patients reaching the endpoint displayed a significantly increased normalized relative expression of miRNA 122-5p, compared to their event-free counterparts. Such a robust association was corroborated by unadjusted Kaplan–Meier survival analyses and, more importantly, confirmed by either univariate or multivariate time-dependent Cox regression analyses, which depicted an independent remarkable 11% increase in the risk of the combined outcome per each 2^−ΔCT^ (×10^3^) increase in the circulating miRNA 122-5p relative expression.

As previously said, rather paradoxically, higher miRNA 122-5p blood concentrations were associated with a higher risk, despite the fact that the whole cohort displayed, on average, lower circulating levels than healthy subjects. Although this finding might appear contradictory, various inferences can be made in the setting of the complexity of factors which characterize the ESKD setting. Among these, the hepatic origin of miRNA 122-5p, which is suggested by the close relationship found with ALT levels, should be taken into consideration. Like aminotransferases, lower cut-off values for miRNA 122-5p might apply to HD patients as compared to the general population [30]; moreover, like miRNA 122-5p, positive associations have previously been described between higher aminotransferase levels and an increased mortality risk in HD patients, despite the fact that lower baseline levels were reported on average compared to healthy individuals [31]. Finally, as previously mentioned, more increased levels of miRNA 122-5p might anticipate incipient and subtle liver injury in the cases of fluid overload and hepatic congestion, thereby reflecting a more severe but still compensated dysfunction of the whole CV system.

Our study has strengths and weaknesses that merit a brief mention. Points of strengths are the multicentric cohort, the systematic collection of events in the framework of a widely acknowledged composite CV endpoint, the robustness of data and follow-up analyses and the high consistency of miRNA 122-5p measurements with those reported by other existing studies. The observational nature of the study probably represents the main weakness, as it does not allow us to clarify the exact biological meaning of the cross-sectional relationships of miRNA 122-5p, its exact role in the context of the high CV risk (causal factor, compensative mechanism or simple epiphenomenon?) and could raise concerns regarding the possible presence of selection bias and residual confounding, that may have hampered our results. Lastly, the sample size and the duration of follow-up, despite being adequate for performing reliable analyses without model overfitting, limited the possibility of running more complex investigations, such as subgroup and exploratory analyses on the single components of the composite endpoint.

## 5. Conclusions

We have demonstrated that altered circulating miRNA 122-5p levels may reflect hepatic and CV injury in uremic patients on chronic HD treatment. Future mechanistic studies are needed to clarify the exact meaning and the biological importance of these observations in the framework of the increased CV risk which characterizes this kind of population. No less important, additional clinical evidence in larger and more heterogeneous HD cohorts is eagerly advocated, to validate the potential utility of measuring circulating miRNA 122-5p levels as a complementary instrument to refine CV risk prediction in such a particular disease setting.

## Figures and Tables

**Figure 1 biomolecules-13-01663-f001:**
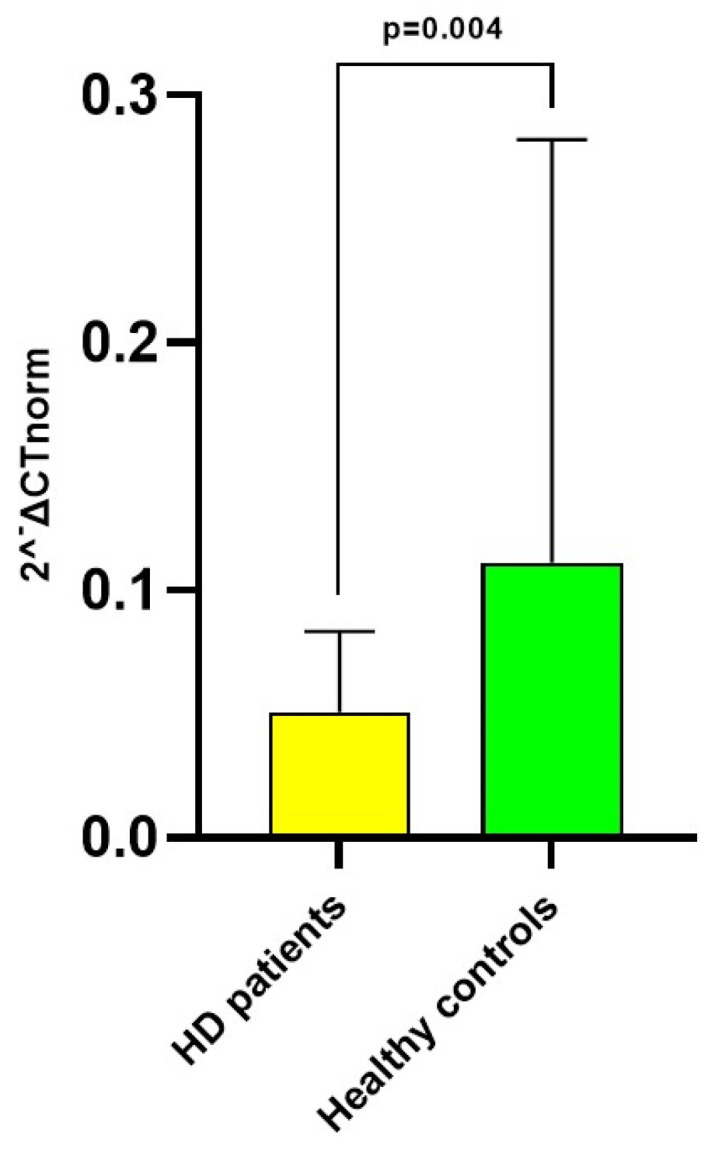
Difference in median circulating miRNA 122-5p (2^−ΔCT^ normalized relative expression) between HD patients and healthy controls.

**Figure 2 biomolecules-13-01663-f002:**
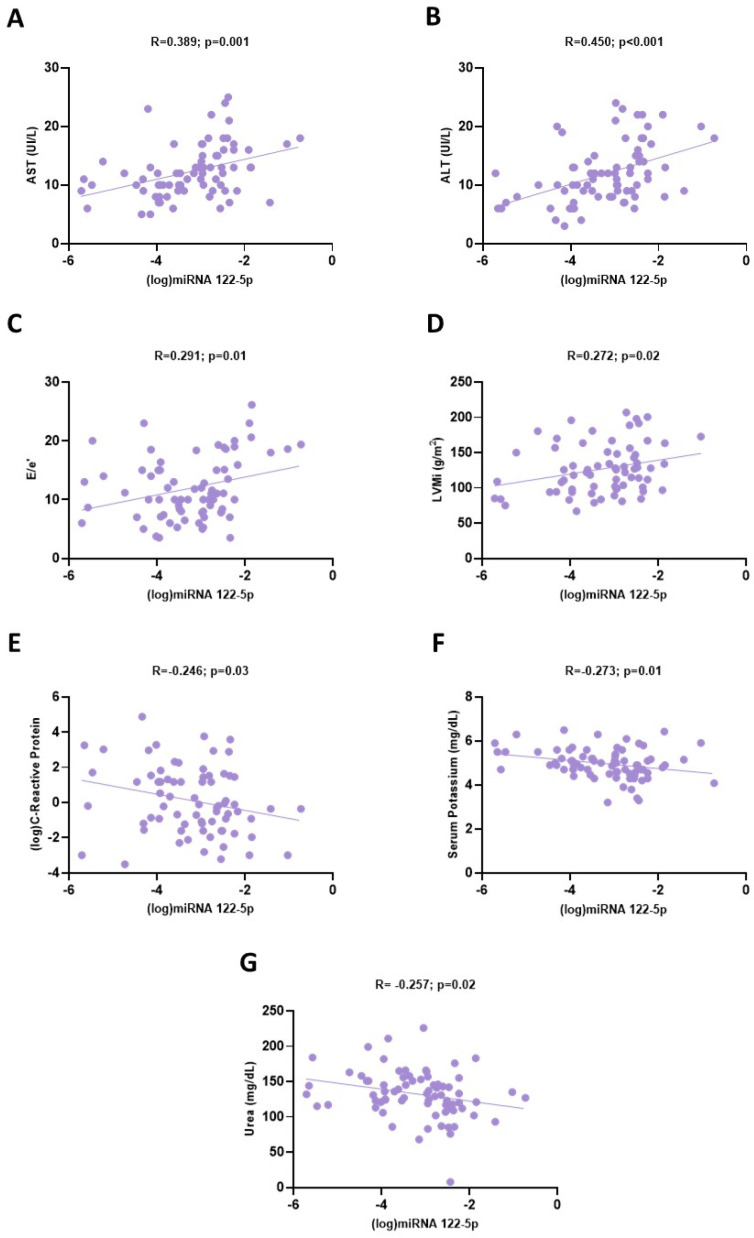
Univariate associations between circulating miRNA 122-5p expression (log) and: (**A**) blood AST values; (**B**) ALT values; (**C**) E/e’; (**D**) LVMi; (**E**) (log)C-reactive protein; (**F**) serum potassium and (**G**) urea levels.

**Figure 3 biomolecules-13-01663-f003:**
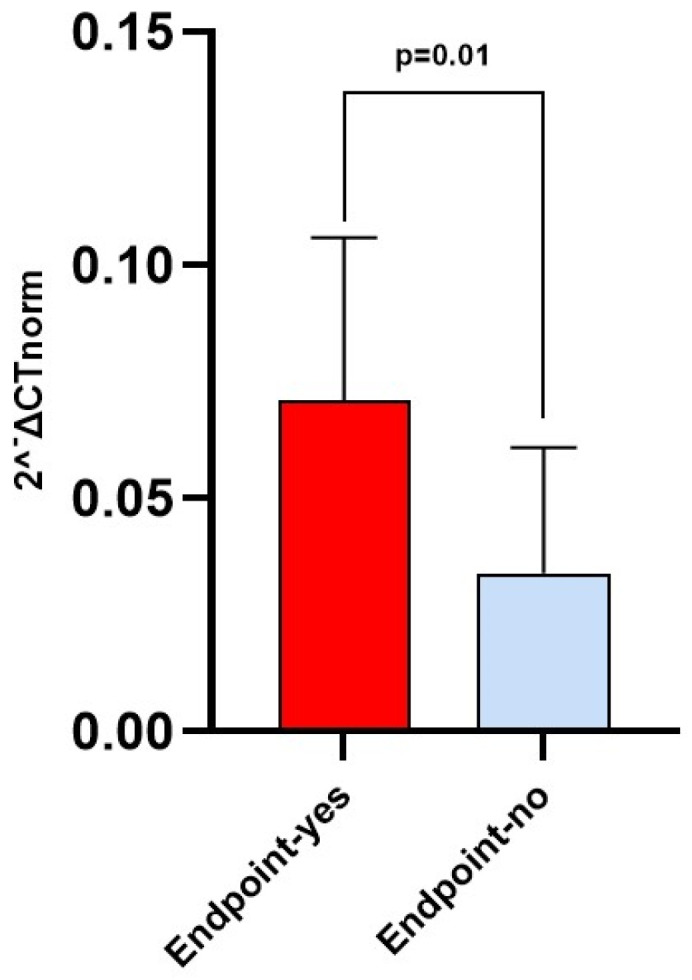
Difference in median circulating miRNA 122-5p relative expression (2^−ΔCT^ normalized relative expression) in patients experiencing the composite endpoint (*n* = 30) as compared to others (*n* = 44).

**Figure 4 biomolecules-13-01663-f004:**
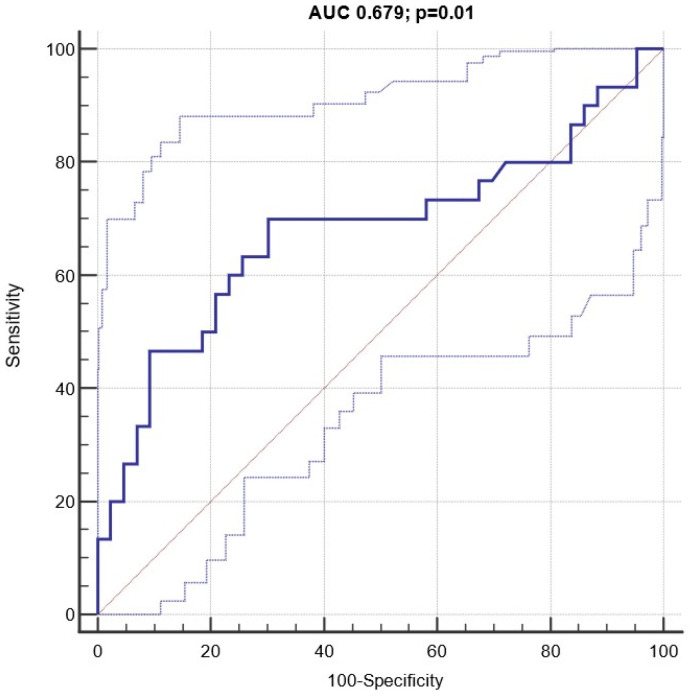
Receiver operating characteristic (ROC) curve (with 95% CI) testing the diagnostic capacity of circulating miRNA 122-5p expression in identifying patients experiencing the composite outcome. The best discriminatory cut-off value (Youden index) of miRNA 122-5p expression was >0.0521 2^−ΔCT^ (sensitivity 70%, 95% CI 50.6–85.3; specificity 69.77%, 95% CI 53.9–82.8).

**Figure 5 biomolecules-13-01663-f005:**
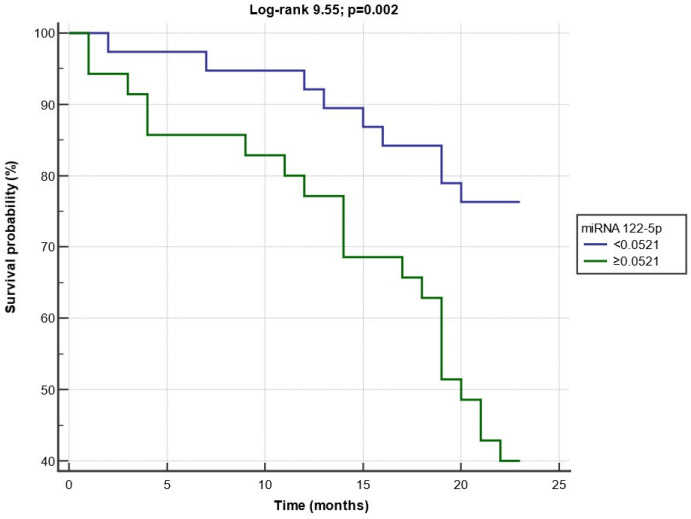
Kaplan–Meier curves of endpoint-free patients with circulating miRNA 122-5p expression below or above the optimal ROC-derived cut-off.

**Table 1 biomolecules-13-01663-t001:** Main clinical characteristics in all HD patients and in individuals categorized according to the occurrence of the composite endpoint. Statistically significant differences between subgroups are highlighted in bold.

	All HD Patients *n* = 74	Endpoint—No *n* = 44	Endpoint—Yes *n* = 30	*p*
Age (years)	68.6 ± 12.3	**66.6 ± 11.9**	**72.5 ± 12.5**	**0.04**
Males *n* (%)	56 (75.7)	33 (75)	23 (76.7)	0.90
Dry weight (kg)	71 ± 15.3	72.1 ± 16.8	71.7 ± 13.2	0.92
BMI (kg/m^2^)	25.4 ± 4.9	26.2 ± 5.2	25.1 ± 3.8	0.76
Waist–hip ratio (cm)	0.94 ± 0.12	0.92 ± 0.23	0.95 ± 0.19	0.69
Type of dialysis				
- Hemodialysis *n* (%)	41 (55.4)	20 (45.4)	21 (70)	0.64
- Hemodiafiltration *n* (%)	33 (44.6)	24 (54.5)	9 (30)	0.06
Kt/V	1.44 ± 0.27	1.46 ± 0.27	1.40 ± 0.28	0.29
History of any CV disease	36 (48.6)	12 (27.3)	18 (60)	0.08
Dialysis vintage (mo.)	35.5 [17–68.2]	35.5 [21.2–66.7]	40 [15.5–71.2]	0.76
Diabetes *n* (%)	20 (27)	11 (25)	9 (30)	0.79
Glycemia (mg/dL)	113 ± 49.4	125 ± 42	140.5 ± 58.2	0.18
Hemoglobin (g/dL)	10.9 ± 1.1	11.1 ± 1.15	10.7 ± 1.1	0.11
SBP (mmHg)	139.6 ± 23.8	141.4 ± 19.4	136.9 ± 29.3	0.42
DBP (mmHg)	71.4 ± 12.2	**73.7 ± 10.2**	**58 ± 14.2**	**0.04**
Pulse pressure (mmHg)	68.1 ± 26.8	**57.5 ± 23.2**	**76.8 ± 25.6**	**0.03**
Serum creatinine (mg/dL)	8.2 [6.9–9.2]	8.9 [7.5–9.9]	8.60 [7–10.64]	0.11
Urea (mg/dL)	133.1 ± 34	132.2 ± 32	134.9 ± 32.8	0.58
Sodium (mg/dL)	136.3 ± 15.2	135.6 ± 19.6	137.4 ± 3.1	0.60
Potassium (mg/dL)	4.97 ± 0.68	5.01 ± 0.65	4.91 ± 0.73	0.52
Phosphate (mg/dL)	4.7 ± 1.29	4.75 ± 1.24	4.65 ± 1.38	0.74
Calcium (mg/dL)	9.23 ± 0.64	9.30 ± 0.69	9.12 ± 0.57	0.26
iPTH (pg/mL)	251.9 [145.9–386]	207 [129–255]	239.8 [124.5–358.9]	0.59
Uric acid (mg/dL)	5.8 ± 1.1	5.8 ± 1.04	5.82 ± 1.15	0.92
Albumin (g/dL)	3.97 ± 0.24	3.99 ± 0.19	3.94 ± 0.31	0.41
CK-MB (UI/L)	21 ± 8.1	**18.8 ± 6.6**	**24.2 ± 9.2**	**0.02**
hs-cTnI (pg/mL)	21.4 [11.6–61.7]	**19.3 [7.7–22.2]**	**23.1 [20.6–53.6]**	**0.04**
ALP (U/L)	79 [62.7–88.5]	79 [62–87]	85.5 [67.5–92]	0.23
Total cholesterol (mg/dL)	147.5 ± 45.1	141.8 ± 28.9	155.8 ± 61.3	0.19
HDL (mg/dL)	41.6 ± 10.1	40.9 ± 9.3	42.5 ± 11.3	0.50
LDL (mg/dL)	83.6 ± 33	**77.9 ± 25.8**	**91.9 ± 30.5**	**0.05**
Triglycerides (mg/dL)	126 [85.5–194.5]	127 [99–173]	140 [90.5–265.7]	0.61
AST (U/L)	12.3 ± 4.4	11.8 ± 3.8	13 ± 5.2	0.57
ALT (U/L)	11.9 ± 5	11.3 ± 4.4	12.8 ± 5.9	0.66
ESR (mm/h)	29 [20–44.7]	33 [25–47]	29 [25.2–45.2]	0.36
C-reactive protein (mg/L)	0.81 [0.23–3.23]	0.30 [0.12–0.52]	0.61 [0.11–0.93]	0.23
Fibrinogen (mg/dL)	421.6 ± 100.9	414.8 ± 60.2	431.6 ± 54	0.58
RBC (*n* × 10^3^)	3.70 ± 0.73	3.71 ± 0.50	3.69 ± 0.99	0.88
WBC (*n* × 10^3^)	7.01 ± 3.30	7.17 ± 2.9	6.77 ± 1.94	0.60
PLT (*n* × 10^3^)	219.2 ± 72.1	216.7 ± 75.6	222.8 ± 67.6	0.72
Serum iron (mg/dL)	70.9 ± 25.2	72.2 ± 25.9	69 ± 24.2	0.59
TSAT (%)	28 [20.2–34.3]	28.2 [22.1–32.1]	28.9 [23.1–34.5]	0.88
Ferritin (mcg/L)	197.9 [103–318]	178.3 [92–251]	258.1 [129.7–341.8]	0.32
miRNA 122-5p (2^−ΔCT^)	0.051 [0.019–0.083]	**0.034 [0.019–0.061]**	**0.071 [0.022–0.106]**	**0.01**

Legend: ALP, alkaline phosphatase; ALT, alanine aminotransferase; AST, aspartate aminotransferase; BMI, body mass index; CK-MB, myocardium specific creatine kinase; DBP, diastolic blood pressure; ESR, erythrocyte sedimentation rate; HDL, high density lipoprotein; hs-cTnI: high-sensitivity c-Troponin I; LDL, low density lipoprotein; PLT, platelet count; iPTH, intact parathormone; RBC, red blood cell count; SBP, systolic blood pressure; TSAT, transferrin saturation; WBC, white blood cell count.

**Table 2 biomolecules-13-01663-t002:** Echocardiography parameters in all HD patients and in individuals categorized according to the occurrence of the composite endpoint. Statistically significant differences between the two subgroups are highlighted in bold.

	All HD Patients *n* = 74	Endpoint—No *n* = 44	Endpoint—Yes *n* = 30	*p*
LAVi (mL/m^2^)	28.3 [21.9–42.8]	**26.3 [18.3–42.9]**	**29.2 [23.6–41.9]**	**0.05**
LAD (cm)	3.7 [3.3–4.1]	3.8 [3.4–4.5]	3.7 [3.2–3.9]	0.37
LVEDVi (mL/m^2^)	51.3 ± 21.9	50.2 ± 20	53.1 ± 25.2	0.17
LVMi (g/m^2^)	131.1 ± 36.8	**128.8 ± 33.2**	**134.8 ± 41.3**	**0.05**
Ejection fraction (%)	57.5 ± 9.1	58.9 ± 7.6	55.3 ± 10.9	0.11
Vmax (m/s)	2.5 [2.2–2.9]	2.44 [2–2.78]	2.49 [1.92–2.89]	0.23
TAPSE (mm)	20 ± 4.8	21 [18–25]	21 [19–26]	0.97
E/e’	13.2 ± 4.2	**10.5 ± 3.6**	**16.2 ± 4.8**	**0.01**
Fractional shortening (%)	37 ± 8.7	37.5 ± 7.9	36.2 ± 10.2	0.60
RAVi (mL/m^2^)	29.5 [14–25.3]	18.1 [11.8–24.3]	20.3 [15.9–24.9]	0.23

Legend: LAVi, left atrial volume index; LAD: left atrial diameter; LVEDVi, left-ventricular end diastolic volume index; LVMi, left ventricular mass index; Vmax: peak aortic valve velocity; TAPSE, tricuspid annular plane excursion; E/e’, early diastolic peak left ventricular inflow velocity (E)/early diastolic peak lateral mitral annular velocity (e’) ratio; RAVi, right atrial volume index.

**Table 3 biomolecules-13-01663-t003:** Univariate and multivariate correlates of miRNA 122-5p expression in HD patients. Statistically significant associations maintained in the fully adjusted multivariate model are highlighted in bold.

	Univariate R/Rho	*p*	Multivariate R^2^ (*p*)	Multivariate β	*p*
Variable			0.39 (<0.001)		
ALT	**0.450**	**<0.001**		**0.333**	**0.02**
E/e’	**0.291**	**0.01**		**0.265**	**0.02**
(log)C-reactive protein	**−0.246**	**0.03**		**−0.219**	**0.04**
LVMi	0.272	0.02		0.168	0.12
Serum potassium	−0.273	0.01		0.127	0.25
AST	0.389	0.001		0.089	0.55
Urea	−0.257	0.02		−0.047	0.67

Legend: ALT, alanine aminotransferase; AST; aspartate aminotransferase; E/e’, early diastolic peak left ventricular inflow velocity (E)/early diastolic peak lateral mitral annular velocity (e’) ratio; LVMi, left ventricular mass index.

**Table 4 biomolecules-13-01663-t004:** Univariate and multivariate Cox regression analyses for variables significantly associated with the composite endpoint. Statistically significant predictors are highlighted in bold.

	Units of Increase	HR	95% CI	χ^2^	*p*
Univariate Analysis					
Pulse pressure	**1 mmHg**	**1.026**	**1.008–1.044**	**8.003**	**0.005**
E/e’	**1U**	**1.155**	**1.081–1.234**	**18.343**	**<0.001**
CK-MB	**1 UI/L**	**1.062**	**1.014–1.112**	**6.618**	**0.01**
miRNA 122-5p	**2^−ΔCT^**	**3.235**	**1.343–4.805**	**4.651**	**0.03**
Age	years	1.036	0.994–1.081	2.754	0.09
LAVi	mL/m^2^	1.016	0.983–1.049	1.137	0.34
LVMi	g/m^2^	1.011	0.996–1.025	2.232	0.14
LDL	mg/dL	1.013	0.999–1.028	1.904	0.11
hs-CTNi	pg/mL	1.006	0.997–1.015	1.301	0.22
Multivariate Analysis					
E/e’	**1U**	**1.085**	**1.028–1.146**	**8.736**	**0.003**
miRNA 30a-5p	**2^−ΔCT^ (×10^3^)**	**1.115**	**1.009–1.232**	**4.576**	**0.03**
CK-MB	1 UI/L	1.007	0.997–1.018	2.008	0.15
Pulse pressure	1 mmHg	1.013	0.988–1.038	1.084	0.29

Legend: CK-MB, myocardium specific creatine kinase; E/e’, early diastolic peak left ventricular inflow velocity (E)/early diastolic peak lateral mitral annular velocity (e’) ratio; hs-cTnI: high-sensitivity c-Troponin I; LAVi, left atrial volume index; LDL, low density lipoprotein; LVMi, left ventricular mass index.

## Data Availability

Raw data from the present study are available from the corresponding author upon reasonable request.

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
