# Peer review of "Circulating miRNA 122-5p Expression Predicts Mortality and Cardiovascular Events in Chronic Hemodialysis Patients: A Multicentric, Pilot, Prospective Study"

_biomolecules, 2023, doi:10.3390/biom13111663_

Round 1
Reviewer 1 Report
Comments and Suggestions for Authors
In this paper, the authors examined the predictive role of circulating miRNA 122-5p expression in hemodialysis patients. The topic is interesting, and the obtained results have practical significance.
Proposed corrective measures:
ü harmonize the way of citing “miRNA”, because in the summary and introduction there are places where it is quoted as “MiRNA”
ü the summary did not introduce the abbreviation for end-stage kidney disease (ESKD)
ü in the description of the study population, the items related to the hemodialysis procedure are not detailed, only from the results we learn that some of the patients were on the hemofiltration regimen. In this sense, it is impossible to consider all patients as a homogeneous population, but separate two subpopulations. Also, the results mention a control group of healthy subjects that was not described at all in the study population section (see Figure 1)
ü it is necessary to describe in detail the autoanalyzers used for measuring laboratory parameters, as well as to specify the reference ranges for the same
ü harmonize the indication of KT/V, kt/V…?
ü include all parameters in the regression analysis, including gender, given that men made up the majority of the study population
Considering that it is a prospective and multicentric study, I think that the work deserves attention and that correction will improve its quality.
Author Response
In this paper, the authors examined the predictive role of circulating miRNA 122-5p expression in hemodialysis patients. The topic is interesting, and the obtained results have practical significance.
Proposed corrective measures:
-harmonize the way of citing “miRNA”, because in the summary and introduction there are places where it is quoted as “MiRNA”
The nomenclature has now been harmonized as suggested by the reviewer
- the summary did not introduce the abbreviation for end-stage kidney disease (ESKD)
The term ESKD has now been removed from the abstract
-in the description of the study population, the items related to the hemodialysis procedure are not detailed, only from the results we learn that some of the patients were on the hemofiltration regimen. In this sense, it is impossible to consider all patients as a homogeneous population, but separate two subpopulations. Also, the results mention a control group of healthy subjects that was not described at all in the study population section (see Figure 1)
We thank the reviewer for pointing out these aspects. The dialysis scheme and modality have now been detailed in the inclusion criteria. Nevertheless, we would like to point out that the exploratory analyses did not uncover significant differences in the main clinical variables recorded (including miRNA 122-5p levels) or in the occurrence of the outcome. Regarding the control group, this consisted of 14 healthy matched individuals (10 male) with no evidence or clinical history of renal, hepatic or CV disease. This description has now been implemented. These subjects served only as controls for miRNA 122-5p measurement, to establish comparable values in a non-ESKD population. Unfortunately, the majority of clinical or laboratory variables pertaining the daily management of ESKD patients were not recent or unavailable in these healthy individuals, thereby making difficult to perform an extensive description of such population (e.g. in a table).
- it is necessary to describe in detail the autoanalyzers used for measuring laboratory parameters, as well as to specify the reference ranges for the same
Detailed information on the lab analyzers used for laboratory measurements have now been provided.
-harmonize the indication of KT/V, kt/V…?
The nomenclature has now been harmonized all over the manuscript as “Kt/V”
- include all parameters in the regression analysis, including gender, given that men made up the majority of the study population
We thank the reviewer for this suggestion. We would point out that gender did not show any correlation with the study endpoint at univariate analyses and the percentage of male individuals among patients experiencing the endpoint or not was almost identical (chi square p=0.90). Given the rate of patients reaching the endpoint during follow-up (40.5%), we believe that the inclusion of non-univariate predictors in multivariate adjustments would end up overfitting the model with any final gain in the predictive value of the analysis.
Reviewer 2 Report
Comments and Suggestions for Authors
The study performed a cohort study on 74 patients undergoing chronic hemodialysis (HD), and found that high miRNA 122-5p levels in HD patients may reflect hepatic and cardiovascular damage. The results are quite interesting. However, some majors issues need to be addressed before the manuscript can be published.
1. The general profile of circulating miRNAs in patients undergoing chronic hemodialysis should be provided.
2. The potential down-pathway of miR-122-5p can be tested in above clinical samples to confirm the conclusion.
Author Response
The study performed a cohort study on 74 patients undergoing chronic hemodialysis (HD), and found that high miRNA 122-5p levels in HD patients may reflect hepatic and cardiovascular damage. The results are quite interesting. However, some majors issues need to be addressed before the manuscript can be published.
- The general profile of circulating miRNAs in patients undergoing chronic hemodialysis should be provided.
We thank the reviewer for pointing out this. Unfortunately, any “general” profile fitting the circulating features of all known miRNAs can be described so far as the large, accumulated evidence on this issue displays conflicting conclusions in such regard. miRNAs can be in principle be freely filtrated by hemodialysis, which would in principle lead to observe lower levels for all miRNAs in HD patients as compared with healthy subjects. For some miRNAs, the lower circulating levels may also reflect, at least in part, systemic or organ-specific downregulation, rather than an excessive dialysis removal. Yet, for some other miRNAs, there is paradoxical evidence of higher circulating levels in dialysis patients as compared to healthy individuals, which may reflect important systemic or organ-specific upregulation as a pathogenic mechanism or as an epiphenomenon of a particular disease or condition associated with renal failure or the dialysis treatment itself (eg. CV disease, inflammation, osteopenia….). The difficulty in defining an univocal profile of circulating miRNAs in HD patients has been acknowledged in the discussion section.
- The potential down-pathway of miR-122-5p can be tested in above clinical samples to confirm the conclusion.
We agree with the reviewer on the opportunity of complementing the findings reported from the present study with targeted experimental investigations testing miRNA regulation directly in blood samples. However, this was not the aim of the present study which was designed as a pilot, proof-of-concept, clinical, observational study to test an ongoing rationale. Again, we agree that the hypothesis generated by our findings strongly deserve to be better interpreted on the basis of experimental studies. This has been acknowledged in the discussion and conclusion sections.
Round 2
Reviewer 1 Report
Comments and Suggestions for Authors
I am glad that the authors took into account all the reviewers' suggestions and I think that the quality of the manuscript has now improved, both in terms of content and technique.
Author Response
We thank the reviewer for the time spent in reviewing our manuscript and for the precious comments
Reviewer 2 Report
Comments and Suggestions for Authors
The manuscript looks OK, though the authors refused to improved it.
Author Response
We are glad to know that the manuscript is now considered suitable for publication in the present form